# Methane Production of Fresh Sainfoin, with or without PEG, and Fresh Alfalfa at Different Stages of Maturity is Similar but the Fermentation End Products Vary

**DOI:** 10.3390/ani9050197

**Published:** 2019-04-26

**Authors:** Pablo José Rufino-Moya, Mireia Blanco, Juan Ramón Bertolín, Margalida Joy

**Affiliations:** Centro de Investigación y Tecnología Agroalimentaria de Aragón (CITA), Instituto Agroalimentario de Aragón–IA2 (CITA-Universidad de Zaragoza), Avda, Montañana 930, 50059 Zaragoza, Spain; pjrufino@cita-aragon.es (P.J.R.-M.); mblanco@aragon.es (M.B.); jrbertolin@cita-aragon.es (J.R.B.)

**Keywords:** polyethylene glycol, gas production, in vitro organic matter degradability, condensed tannins, ammonia, volatile fatty acid, in vitro assay

## Abstract

**Simple Summary:**

In the last years, there has been increasing interest in the use of forages containing condensed tannins (CT) in ruminant nutrition. Condensed tannins can reduce the methane emissions and the ruminal degradation of protein, improving the animal performances to different extents depending on the source and dose of CT. In vitro fermentation of sainfoin has not been studied in fresh forage. The effect of CT can be studied in comparison with a similar CT-free forage or using polyethylene glycol (PEG), which is a tannin-blocking agent. The maturity stage influences the chemical composition to a different degree depending on the legume species, and can affect the content and fractions of CT. The aims of this trial were to compare the fermentation parameters of sainfoin with or without PEG, to detect the differences due to CT, at different maturity stages (vegetative, start-flowering, and end-flowering) and compare them with the fermentation parameters of alfalfa. The main results were that sainfoin had greater in vitro organic matter degradability (IVOMD) and lower ammonia and acetic:propionic ratio than alfalfa. Sainfoin CT had effect on ammonia and individual fatty acid proportions. In conclusion, fermentation end products were affected both by the chemical composition and CT contents.

**Abstract:**

Alfalfa and sainfoin are high-quality forages with different condensed tannins (CT) content, which can be affected by the stage of maturity. To study the effects of CT on fermentation parameters, three substrates (alfalfa, sainfoin, and sainfoin+PEG) at three stages of maturity were in vitro incubated for 72 h. Sainfoin had greater total polyphenol and CT contents than alfalfa. As maturity advanced, CT contents in sainfoin decreased (*p* < 0.05), except for the protein-bound CT fraction (*p* > 0.05). The total gas and methane production was affected neither by the substrate nor by the stage of maturity (*p* > 0.05). Overall, sainfoin and sainfoin+PEG had greater in vitro organic matter degradability (IVOMD) than alfalfa (*p* < 0.05). Alfalfa and sainfoin+PEG presented higher ammonia content than sainfoin (*p* < 0.001). Total volatile fatty acid (VFA) production was only affected by the stage of maturity (*p* < 0.05), and the individual VFA proportions were affected by the substrate and the stage of maturity (*p* < 0.001). In conclusion, alfalfa and sainfoin only differed in the IVOMD and the fermentation end products. Moreover, CT reduced ammonia production and the ratio methane: VFA, but the IVOMD was reduced only in the vegetative stage.

## 1. Introduction

There is increasing interest in legume-based forage production systems because of their low reliance on fertilizer nitrogen, potential environmental benefits, and high protein content that contribute to low-input and sustainable livestock production systems [1]. Alfalfa (*Medicago sativa L*.) and sainfoin (*Onobrychis viciifolia Scop*.) are two pluriannual legumes widely grown in the Mediterranean area, presenting high forage productive capacity, high nutritional value, and restorative action for soil fertility [2]. However, alfalfa has a low content of polyphenols and is considered virtually free of condensed tannins (CTs), whereas sainfoin presents a high content of polyphenols and a medium to high content of CTs [3,4,5]. 

Alfalfa is usually fed as hay to ruminants mainly to avoid bloat, although continuous grazing is possible without bloat incidence both in sheep [6] and growing cattle [7]. Thus, the majority of studies that compared the ruminal fermentation of alfalfa and sainfoin have been performed using hays [3,4,8]. The differences between alfalfa and sainfoin have been ascribed to differences in the chemical compositions [4], but also to the presence of CTs [8]. To clarify whether the differences between alfalfa and sainfoin are only due to CTs, polyethylene glycol (PEG) must be added as a tannin-blocking agent [8]. To the best of our knowledge, the ruminal fermentation of both species was compared in fresh forages only by McMahon et al. [9] and Chung et al. [10]. Depending on their content, characteristics, and properties [11], CTs from sainfoin alter both the breakdown of protein in the rumen to ammonia and methane, gas and the production of total volatile fatty acids (VFAs) [3,4,12]. These, in turn, are associated with the variety, stage of maturity, and environmental factors [5,13,14].

The objectives of this trial were to compare the in vitro fermentation of alfalfa and sainfoin at three stages of maturity and to clarify whether the differences between both legumes were due to the CTs of sainfoin using PEG.

## 2. Materials and Methods 

### 2.1. Experimental Design

Three substrates (alfalfa, sainfoin, and sainfoin+PEG) at three stages of maturity (vegetative, start-flowering, and end-flowering) were used to evaluate the effect of sainfoin CTs on in vitro fermentation. Alfalfa was used as a tannin-free legume.

### 2.2. Animal and Diets

#### 2.2.1. Forages, Crop Management, and Harvest

The experiment was performed at the CITA Research Institute at Zaragoza (41°42′ N, 0°49′ W), altitude 216 m a.s.l., located in Ebro Valley, north-eastern Spain. The silt loam soil at the site had pH 8.1 and 1.81% organic matter and contained 16% clay, 53.5% silt, and 30.5% sand. Alfalfa and sainfoin were cultivated and managed as described by Lobón et al. [15]. Samples of forage were collected fortnightly from 14 April to 22 September 2015. The stage of maturity of the sainfoin and alfalfa was classified into vegetative, start-flowering, and end-flowering according to Borreani et al. [16] and Kalu and Fick [17], respectively. In each sampling, 10 samples per legume were obtained from 0.25 m^2^ areas randomly allocated in the plot. These samples were mixed homogenously, and a part of the sample was separated manually into stems, leaves, and flowers to study their respective percentages. Another part of the samples was dried at 60 °C for 48 h for chemical analyses, and the rest of the sample was freeze-dried in a Genesis Freeze Dryer 25 (Hucoa Erlöss, SA/Thermo Fisher Scientific, Madrid, Spain) for polyphenol and CT analyses and in vitro fermentation assays. Samples for the chemical analyses were ground and sieved through a 1-mm screen (Rotary Mill, ZM200 Retsch, Hann, Germany), and a part of the samples was ground and sieved through a 0.2-mm screen for crude protein (CP), polyphenol, and CT determination. All the samples were stored at −20 °C in total darkness. 

The number of samples of vegetative, start-flowering and end-flowering stages of each forage to study the in vitro fermentation were chosen in concordance with the representativeness of the plant development. The vegetative stage was the most frequent stage (55%), followed by end-flowering (27%) and start-flowering (18%) stages. Three samples of the vegetative stage and two samples of the start- and end-flowering of each legume species were studied.

#### 2.2.2. Animals and Sampling of Ruminal Digesta

The procedures used in the trial followed the Spanish guidelines for experimental animal protection (RD 53/2013) and were approved by the Institutional Animal Care and Use Committee of the Research Centre (Procedure number 2011-05). Rasa Aragonesa wethers (n = 4; Live weight: 65 ± 2.1 kg) were used as donors of ruminal content. The management of the rumen-cannulated wethers and the sampling of the ruminal digesta was made as reported in Rufino-Moya et al. [18]. Briefly, wethers were fed a diet composed by alfalfa hay (70%) and barley grain (30%) at energy maintenance level. Before morning feeding, ruminal digesta from each wether was collected into a prewarmed insulated thermo, individually strained through four layers of cheesecloth and homogenized. Rumen fluid from the four wethers was mixed (pH: 6.76 ± 0.099), and a buffer solution was added in a proportion of 1:2 (*v*/*v*) based on the protocol of Menke and Steingass [19]

#### 2.2.3. In Vitro Gas Production Technique and Sampling

The production of gas was measured with the Ankom system (Ankom Technology Corporation, Fairport, NY, USA), which had 310 mL capacity bottles fitted with pressure and temperature sensors. The valve open time was one second, the threshold for gas release was 5 PSI and the bottles were not shaken. Five hundred mg of freeze-dried substrate (alfalfa, sainfoin, or sainfoin+PEG) were incubated with 60 mL of buffered solution:rumen fluid (2:1 *v*/*v*) in a water bath (at 39 °C) for 72 h. To make the sainfoin+PEG samples, PEG-4000 (Merck, Darmstadt, Germany) was added to the buffered rumen fluid at a concentration of 2.3 g/L [12]. Four runs were conducted on four separate days, and each sample was incubated in duplicate in each run. Gas production and corrected with the blanks (two bottles without substrate were included in each run).

After 72 h of incubation, the bottles were placed in ice to stop fermentation (5–10 min), and then tempered (at room temperature for 10–15 min). A sample of the gas produced was transferred (at atmospheric pressure) with a syringe attached to a manometer into a Vacutainer® tube to determine CH_4_ and conserved at 4 °C until analysis. At the end of gas sampling, the pH was measured immediately with a microPH 2002 (Crison Instruments S.A., Barcelona, Spain). The sampling to determine ammonia (NH_3_-N) content and VFA were carried out as reported in Rufino-Moya et al. [18]. Briefly, to determine the ammonia content, 2.5 mL of liquid was mixed with 2.5 mL HCl 0.1 N. For VFA determination, 0.5 mL of the liquid was added to 0.5 mL of deproteinizing solution and 1 mL of distilled water. Tubes with samples for determination of ammonia and VFAs were stored at −20 °C until future analyses. The entire incubated sample was filtered through a preweighed bag (50 μm; Ankom) to estimate the in vitro organic matter degradability (IVOMD). Briefly, the bags were sealed, washed, dried at 103 °C for 48 h, and finally, placed in a muffle at 550 °C to obtain the ashes. The organic matter of the bag content was obtained as DM-ashes, and the IVOMD was calculated.

### 2.3. Analytical Methods

#### 2.3.1. Chemical Composition

All the analyses of the chemical composition were analyzed as reported in Rufino-Moya et al. [18] according to official methods. Briefly, AOAC methods were used to determine the contents of dry matter (index no. 934.01), ash (index no. 942.05), and CP (index no. 968.06) [20]. Neutral detergent fiber (NDFom), acid detergent fiber (ADFom), and lignin (sa) contents were determined according to the method described by Van Soest et al. [21] using the Ankom 200/220 fiber analyzer (Ankom Technology Corporation). The NDFom was assayed with a heat stable amylase. The lignin (sa) was analyzed in ADF residues by the solubilization of cellulose with sulfuric acid. All the values were corrected for ash-free content. The ether extract (EE) was determined following the approved procedure Am 5-04 [22] using an XT10 Ankom extractor (Ankom Technology Corporation). The nonstructural carbohydrates (NSC) were calculated as NSC=1000−CP−EE−NDF−ash, according to Guglielmelli et al. [3]. 

The content of total polyphenol (TP) and the fractions of CT were determined as described in Rufino-Moya et al. [18]. Briefly, the TP were extracted using the method of Makkar [23] and were quantified following the method of Julkunen-Tiitto [24]. The extractable CT (ECT), protein-bound CT (PBCT), and fiber-bound CT (FBCT) were extracted and fractioned following the method of Terrill et al. [25] and quantified by the colorimetric HCl-butanol method described by Grabber et al. [26]. The standard used for the quantification was extracted and purified from sainfoin using the method described by Wolfe et al. [27]. An Heλios β spectrophotometer was used to measure the samples and standard calibration at 550 nm.

#### 2.3.2. Determination of Parameters of the In Vitro Gas Production Technique

Gas production, recorded hourly for 72 h, was used to estimate the parameters of the kinetics of fermentation adjusting the gas produced to the model described by France et al. [28]:$$P=A\times \left(1-e^{-ct}\right)$$
where *P* is the cumulative gas production (mL) at time *t* (h), *A* is the potential gas production (mL), and *c* is the rate of gas production (h^−1^).

An HP-4890 (Agilent, St. Clara, CA, USA) gas chromatograph (GC) equipped with a flame ionization detector (FID) and a TG-BOND Q+ capillary column (30 m × 0.32 mm id × 10 µm film thickness, Thermo Scientific, Waltham, MA, USA) was used to determine CH_4_. Helium was used as the carrier gas at a flow rate of 1 mL/min. The oven temperature was maintained at 100 °C (isothermal program). The splitless injection volume was 200 µL. Methane identification was based on the retention time compared with the standard. The methane concentration was calculated from the peak concentration:area ratio using the peak area generated from standard gas as the reference (CH_4_; 99.995% purity [C45], Carburos Metálicos, Barcelona, Spain).

The content of ammonia was measured in Epoch microplate Spectrophotometer (BioTek Instruments, Inc., Winooski, VT, USA) using a colorimetric method described by Chaney and Marbach [29]. 

A Bruker Scion 460 GC (Bruker, Billerica, MA, USA) equipped with CP-8400 autosampler, FID and a BR-SWax capillary column (30 m × 0.25 mm i.d. × 0.25 µm film thickness, Bruker) was used to determine the concentration of acetic, propionic, iso-butyric, butyric, iso-valeric, and valeric acids. Helium was the carrier gas (flow rate of 1 mL/min). The oven temperature program was 100 °C, followed by a 6 °C/min increase to 160 °C. The injection volume was 1 µL at a split ratio of 1:50. The individual VFAs were identified based on retention time comparisons with commercially available standards of acetic, propionic, iso-butyric, butyric, iso-valeric, valeric, and 4-methyl-valeric acids at ≥99% purity (Sigma-Aldrich, St. Louis, MO, USA).

### 2.4. Statistical Analyses

Data were analyzed using SAS v. 9.3 (SAS Inst. Inc., Cary, NC, USA). The fermentation kinetic parameters (*A* and *c*) were estimated using a nonlinear regression model (NLIN procedure). The contents of secondary compounds were analyzed using the GLM procedure with the substrate, the stage of maturity and its interaction as fixed effects. Total gas and CH_4_ production, *A*, *c*, IVOMD and the fermentation end products were analyzed using mixed models considering the substrate, the stage of maturity and its interaction as fixed effects and the run as random effect. If the interaction was not significant, it was deleted from the model and the analysis was repeated. The least square means, their associated standard errors and the differences were obtained. Pearson correlation coefficients between variables were calculated using the CORR procedure. For the entire test, the effects were considered a significant probability at a value of *p* < 0.05 or a trend at *p* = 0.05.

## 3. Results

### 3.1. Chemical Composition

The chemical composition and the percentage of stems, leaves, and flowers of both legume species at the three stages of maturity are shown in Table 1. On average, alfalfa and sainfoin had similar ADFom (231 g/kg DM), CP (198 g/kg DM), EE (15 g/kg DM), and NSC (335 g/kg DM) contents. Alfalfa, however, had higher ash and NDFom contents and a lower lignin (sa) content than sainfoin. For both forages, NDFom and ADFom content increased as the stage of maturity progressed, whereas the CP content decreased. As the forage matured, the lignin (sa) content increased only in alfalfa whereas the contents of EE and NSC decreased only in sainfoin.

Alfalfa and sainfoin had similar proportions of leaves (51.9 vs. 53.6%). However, alfalfa had a greater proportion of stems (45.2 vs. 39.7%) and a lower proportion of flowers (2.8 vs. 6.7%) than sainfoin. Regarding the stage of maturity, the proportion of stems and flowers increased, whereas the proportion of leaves decreased as the stage of maturity advanced.

### 3.2. Contents of Total Polyphenols and Condensed Tannins

The content of total polyphenols and the total (TCT) and fractions of CT were affected by the interaction between the legume species and the stage of maturity (*p* < 0.05) (Figure 1). Alfalfa presented steady contents of total polyphenols, TCTs, ECTs, PBCTs, and FBCTs, which were lower than those of sainfoin (*p* < 0.001) regardless of the stage of maturity. The contents of polyphenols, TCTs, ECTs, and FBCTs decreased as maturity advanced (*p* < 0.05).

### 3.3. In Vitro Fermentation

The pH was affected by the interaction between the substrate and the stage of maturity (*p* < 0.01; Table 2). Alfalfa had greater pH than sainfoin and sainfoin+PEG (*p* < 0.05). Sainfoin and sainfoin+PEG were affected similarly by the stage of maturity (*p* < 0.001), with the greatest pH value in the vegetative stage (*p* < 0.05) Total gas and CH_4_ production were affected neither by the substrate nor by the stage of maturity (*p* > 0.05; Table 2). However, the interaction between the substrate and the stage of maturity affected *A* (*p* < 0.001) and c (p = 0.05). Alfalfa showed lower *A* values at start-and end-flowering stages (*p* < 0.001) and greater *c* in the vegetative and start-flowering stages (*p* < 0.05) than sainfoin and sainfoin+PEG (*p* < 0.05). Regarding the effect stage of maturity, sainfoin and sainfoin+PEG had the lowest *A* values in the vegetative stage (*p* < 0.05). As the stage of maturity progressed, *c* increased in sainfoin and sainfoin+PEG substrates (*p* < 0.05).

The IVOMD was also affected by the interaction between the substrate and the stage of maturity (*p* < 0.001; Table 2). Alfalfa had lower IVOMD than both sainfoin substrates in the start-flowering and end-flowering stages (*p* < 0.05), whereas sainfoin+PEG had greater IVOMD than alfalfa and sainfoin in the vegetative stage (*p* < 0.05). In alfalfa, the IVOMD decreased as the stage of maturity advanced (*p* < 0.05). The sainfoin and sainfoin+PEG showed the greatest IVOMD in the start-flowering stage (*p* < 0.05). The IVOMD was correlated with *A* (r = 0.60; *P* < 0.01) and with the total VFA production (r = 0.51; *p* < 0.05).

The NH_3_-N content was only affected by the substrate (*p* < 0.001); sainfoin produced a lower NH_3_-N concentration than alfalfa and sainfoin+PEG (Table 2). In contrast, total VFA production was only affected by the stage of maturity (*p* < 0.05), the start-flowering stage presented greater VFA production than the rest of the stages (105, 99, and 100 mmol/L for start-flowering, vegetative, and end-flowering, respectively). Regarding the individual VFAs, alfalfa had a lower acetic acid proportion and greater proportions of the rest of the individual VFAs than sainfoin (*p* < 0.001). When comparing both sainfoin substrates, sainfoin had a greater acetic acid proportion and lower proportions of the rest of the VFAs than sainfoin+PEG (*p* < 0.001). Sainfoin presented the greatest C_2_:C_3_ ratio, followed by sainfoin+PEG and alfalfa, which had the lowest ratio (*p* < 0.001). Regarding the effect of the stage of maturity, the vegetative stage had a lower proportion of acetic acid and greater proportions of propionic, iso-butyric, and iso-valeric acid than the rest of stages of maturity (*p* < 0.001). The vegetative stage had a lower C_2_:C_3_ ratio than the other stages (*p* < 0.001). The CH_4_:VFA ratio was only affected by the substrate (*p* = 0.01); sainfoin+PEG presented a greater CH_4_:VFA ratio than the other substrates (*p* < 0.05).

## 4. Discussion

In Mediterranean areas, there is an increasing interest to reintroduce forage-based systems in ovine production to ensure the viability and sustainability of the farms. Legumes are especially advisable due to their nutritional quality for ruminants and environmental benefits [1]. Moreover, the presence of CT in some legumes may decrease CH_4_ production and improve the performance of ovine to different extents depending on the source and dose of CT. As these legumes are usually fed conserved, either as silage or as hay, there is scarce information on the fermentation parameters when alfalfa and sainfoin are offered fresh. Previous experiments showed differences between the fermentation parameters of alfalfa and sainfoin hay and silage [4,30], however, it is not clear if the differences were due to the chemical composition, the presence of CT in sainfoin or both. With the present study, the fermentation parameters of alfalfa and sainfoin, with or without PEG, in fresh at different stages of maturity were compared to try to clarify the origin of the differences in fermentation. The use of gas production technique is a good tool to evaluate the effect of CT on fermentation parameters, but the fermentation is influenced by the time of incubation, species of the animal donor and the diet [31,32]. Furthermore, the effects of sainfoin CT vary according the variety, harvest time, and cultivation site [13,14], making it difficult to compare the results with other studies. In the present study, the content of total CT and their fractions were analyzed, however, the chemical characteristics of CT (molecular weight, degree of polymerization, prodelphinidin/procyanidin ratio, cis/trans ratio, etc.) were not evaluated.

### 4.1. Effect of the Substrate

References showed noticeable variability in the chemical composition of alfalfa and sainfoin among studies, which is related to the stage at harvest, leaf:stem ratio, soil characteristics, weather conditions, and the cultivars utilized [5,13,14]. In the present study, the similar CP and ADF contents and different NDF contents of alfalfa and sainfoin agree partially with the results reported by Chung et al. [10], who observed similar NDF, ADF and CP contents in fresh alfalfa and sainfoin at the late vegetative stage. However, at the early vegetative stage, the same authors observed greater NDF and ADF contents in alfalfa than sainfoin and similar CP contents.

Alfalfa has a low content of total polyphenols and is considered a CT-free legume [5], although it may present very low CT content in the seed coats. Therefore, the present results related to the presence of CTs agree with previous studies that analyzed both legumes offered fresh [33,34] or as preserved forages [4,30].

The pH did not negatively affect the fermentation environment because the values were within the range of 6.2 to 6.8 and these values ensure a favorable environment for the activity of cellulolytic bacteria [35]. The inclusion of PEG in the current experiment did not affect pH, as reported in fresh sainfoin [12] and sainfoin hay [8]. Regarding gas production, the similar production of alfalfa and sainfoin agrees with the similar gas production observed in alfalfa and sainfoin leaves incubated in Rusitec units [9] and alfalfa and sainfoin silages [30]. However, when alfalfa and sainfoin hays were studied, differences in gas production were reported [3,4,8]. The inconsistency of the results of the type of substrate on gas production might be related to the differences in chemical composition, the different characteristics of CTs and of type and settings of the in vitro assay [14]. In that sense, the similar gas production between sainfoin and sainfoin+PEG was unexpected because previous experiments reported increases in gas production when PEG was added to fresh sainfoin [12,14]. According to Azuhnwi et al. [13], the inclusion of PEG increased gas production by 2.7 to 9.6%, depending on the sainfoin variety, site, and stage at harvest. In the current experiment, the inclusion of PEG slightly increased the gas production, although not statistically significantly.

The similar CH_4_ production of alfalfa and sainfoin recorded in the present study agrees with results reported using fresh forages [10] and silages [30]. However, the inclusion of extracts from sainfoin accessions in alfalfa decreased CH_4_ production, but the effect was greatly dependent on the accession and the dose of inclusion [11]. Moderate CT content may have beneficial effects reducing rumen CH_4_ emission production [36]. The action of CT on methanogenesis can be attributed to indirect effects via reduced hydrogen production (and presumably reduced forage digestibility) and via direct inhibitory effects on methanogens [34]. Regarding the effect of PEG, the inclusion of PEG in sainfoin did not affect CH_4_ production in previous studies [12,37] as in the current experiment. The structural features of condensed tannins affect in vitro CH_4_ production, which may be linked to the interaction of CTs with dietary substrate or microbial cells [11,38]. Therefore, the type of CT and dose present in the current experiment might not be sufficient to modify CH_4_ production.

The reduced *A* in alfalfa compared with sainfoin and sainfoin+PEG in the start- and end-flowering stages can be related to the higher fiber fraction, as reported by Guglielmelli et al. [3]. In the current experiment, the presence of CTs in sainfoin had no effect on *A*, as reported by Calabrò et al. [8]. The higher *c* in alfalfa when compared to sainfoin agrees with the results reported by Hatew et al. [11], although the effect on *c* depends on the types and concentrations of sainfoin CTs.

The lower IVOMD of alfalfa, when compared to sainfoin and sainfoin+PEG, was also reported using fresh forage estimated in situ [10] and in vitro [39] and could be related to the greater fiber fraction. The increased IVOMD in sainfoin+PEG with respect to sainfoin at the vegetative stage could be related to the blockage of CTs by the PEG. However, Theodoridou et al. [12] reported no effect of the inclusion of PEG in sainfoin on IVOMD studied at 24 h, regardless of the stage of maturity. The discrepancy between studies could be related to the content, characteristics and structures of CTs, which depends on the botanical species and variety of the source [13,14].

The NH_3_-N contents recorded in the present study are in line with most similar studies that compared alfalfa and sainfoin [3,4,12]. The reduced NH_3_-N concentration in sainfoin with respect to alfalfa and sainfoin+PEG confirmed the inhibition elicited by CTs in the ruminal degradation of dietary proteins due to the formation of complexes CT-protein at ruminal pH [40]. In contrast, the effect of CTs on total VFA production is not clear. Some studies reported a lower total VFA production in sainfoin silage than in alfalfa silage [30], and the inclusion of different doses of extracted accession of sainfoin in alfalfa decreased or maintained the total VFA production, depending on the accession [11]. In the current experiment, the total VFA production was not affected by the legume species, as observed by other authors [4,10]. The inclusion of PEG did not affect total VFA production in the current experiment as reported for sainfoin hay [8,37], which is contrary to the increase of total VFA production observed by Hatew et al. [14]. The differences between the studies could be due to the time of incubation, species of the animal donor, chemical structure, and biological activity of CTs [14,31].

Generally, the presence of CT from sainfoin leads to an increase in the propionic acid proportion and a reduction in the C_2_:C_3_ ratio [10,11,38]. However, the effect on each individual VFA proportion is variable due to the type of substrate, types, and contents of CTs and length of incubation period or the donor animal [11,31]. In the present study, sainfoin had a greater acetic acid proportion than alfalfa, which was similar to results from Guglielmelli et al. [3] using hays and Grosse-Brinkhaus et al. [30] using silages. The C_2_:C_3_ ratio recorded in the present study was greater in sainfoin than in alfalfa and sainfoin+PEG, which is in contrast with other studies that did not observe effects of the type of substrate or the addition of PEG [3,12,37]. Sainfoin had lower valeric acid and branched-chain VFA proportions than alfalfa and sainfoin+PEG because of the presence of CTs [4,30,38]. Condensed tannins reduce the proportions of branched-chain VFAs due to reduce protein degradation in the rumen because these VFA are products of the breakdown of the carbon skeleton of amino acids during rumen fermentation [41].

### 4.2. Effect of the Stage of Maturity

The decrease in CP content and the concomitant increase in the cell wall (NDFom and ADFom) content as the stage of maturity progressed in both forages is a result of the decrease of the proportion of leaves to stems and the increase of lignified tissues [10]. The steady lignin (sa) content in sainfoin during the development of plants can be due to some interference between this compound and CTs during analysis, as reported by Guglielmelli et al. [3].

In the current experiment, the TP and TCT contents were affected by the stage of maturity, as reported in previous studies [3,5,42], although the magnitude of the effect varied among the studies. Regarding the CT fractions, there is little information about the influence of the stage of maturity. As in the present study, Chung et al. [10] observed a reduction of the ECT fraction in the end-flowering stage with respect to the vegetative stage in sainfoin. However, Jin et al. [43] observed greater TCT, ECT, and PBCT contents in *Dalea purpurea* at flowering than at the vegetative stage due to the high percentage of flowers, which are very rich in CTs [44]. From a physiological point of view, the reduction in the secondary compound concentrations as maturity advances could be due to a sort of dilution as a consequence of the growth and expansion of plant cells [45] and to the decrease in the leaf: stem ratio as a consequence of the reduction of leaves, which are rich in CTs [44].

The lack of an effect of the stage of maturity on pH values in alfalfa agrees with the results reported by Chung et al. [10], but the reduction of pH in sainfoin at both flowering stages disagrees with the abovementioned study. In this sense, the stage of maturity had no effect on pH when sainfoin hay was incubated [3,8]. More studies considering the stage of maturity, the chemical composition and the presence of secondary compounds in alfalfa and sainfoin must be performed to discern the importance of these factors on the ruminal pH.

Previous studies reported a reduction of in vitro gas and CH_4_ production as the stage of maturity advanced, associated with the chemical composition and the CT content [12,46]. However, the stage of maturity had no effect on gas and CH_4_ production in the current experiment, which is in agreement with in vitro [3] and in vivo [10] experiments. The similar chemical composition observed between stages of maturity, the low biological activity of CT and the interactions between nutritive components and antinutritional factors could be responsible for the similar gas and CH_4_ production [3].

As expected, the IVOMD of alfalfa decreased as the fiber fraction increased with maturity [47]. In contrast, sainfoin, with or without PEG, presented considerably high IVOMD at the start-flowering stage with respect to the rest of the stages in agreement with Theodoridou et al. [12]. These results could be due to the different biological activity of CTs at vegetative and start-flowering stages [5], because the chemical composition and CT contents were similar in both stages.

The advancing of the stage of maturity tended to reduce NH_3_-N production as a consequence of the decrease of CP content in concordance with several studies carried out in vitro [3,12] and in vivo [10,42]. Studies concerning the effect of the stage of maturity on the total production of VFAs and their proportions show discrepancies. The chemical composition of the substrates, the length of the incubation period, and the inoculum donor animal are determinant factors that can influence VFA production and proportions [31]. In the current experiment, the start-flowering stage presented the highest total VFA production, in concordance with the highest IVOMD observed. However, Theodoridou et al. [12] studied the effect of the stage of maturity of fresh sainfoin with a similar CT content in a 24 h in vitro assay and did not find an effect on the total VFA production.

In relation to the proportion of individual VFAs, the effect of the stage of maturity on these parameters has been reported in vitro and in vivo in previous studies [3,10], but the results are not consistent. In the current experiment, as maturity advances, there is an increase in acetic acid and a decrease in propionic acid proportions, thus increasing the C_2_:C_3_ ratio due probably to increase of fiber and reduction of CT content [42]. The reduction of the proportion of iso-butyric and iso-valeric acids in the start- and end-flowering stages and valeric acid at the start-flowering stage in comparison with the vegetative stage might be explained by the decrease in CP content, because they are products of the breakdown of the carbon skeleton of amino acids during rumen fermentation, as the maturity of the forage advanced [10,47].

## 5. Conclusions

In conclusion, sainfoin might be an alternative to alfalfa due to the high IVOMD and the potential protection against ruminal protein degradation, according to the results of ammonia content, branched-chain VFAs, and valeric acid proportion from sainfoin in vitro. The effect of the stage of maturity was less than expected, probably due to the high quality of the forages. It is required to study the effects of the type of substrate and stage of maturity on animal performance to recommend the best stage of maturity to cut sainfoin and alfalfa.

## Figures and Tables

**Figure 1 animals-09-00197-f001:**
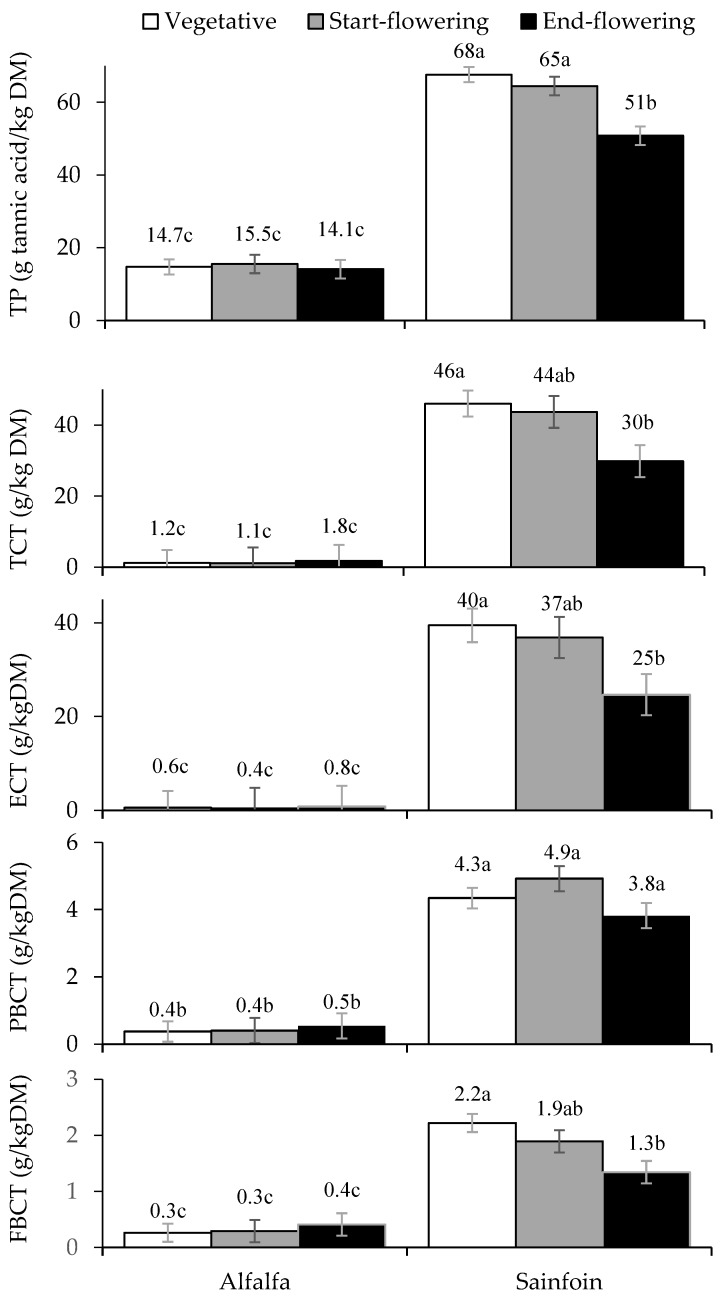
Effect of the species and the stage of maturity on the contents of total polyphenols (TP), total condensed tannins (TCT), extractable CT (ECT), protein-bound CT (PBCT), and fiber-bound (FBCT). Within a parameter, means with different letter differ at *p* < 0.05. (Within each species: n = 3 for the vegetative stage, n = 2 for the start-flowering, and n = 2 for the end-flowering stages.)

**Table 1 animals-09-00197-t001:** Chemical composition and plant components of alfalfa and sainfoin at three stages of maturity.

	Alfalfa			Sainfoin		
Item	Vegetative	Start-Flowering	End-Flowering	Vegetative	Start-Flowering	End-Flowering
Chemical composition						
Dry Matter (DM) (g/kg)	249	261	333	241	262	224
Ash (g/kg DM)	103	111	98	83	113	82
CP ^1^ (g/kg DM)	227	207	166	204	201	181
NDFom ^2^ (g/kg DM)	336	352	405	313	324	394
ADFom ^3^ (g/kg DM)	201	230	276	199	213	264
Lignin (sa), (g/kg DM)	45	53	66	78	76	76
Ether extract (g/kg DM)	18	10	16	22	15	11
NSC ^4^ (g/kg DM)	317	320	316	379	347	332
Plant components (%)						
Leaves	59.3	53.7	42.7	66.3	56.0	38.5
Stems	40.7	43.7	51.3	33.7	40.8	44.6
Flowers	0.0	2.5	5.9	0.0	3.2	16.8

^1^ crude protein; ^2^ Neutral detergent fiber; ^3^ acid detergent fiber; ^4^ nonstructural carbohydrates.

**Table 2 animals-09-00197-t002:** Effect of the substrate (S) and the stage of maturity ^1^ (SM) on gas and methane production (CH_4_), potential gas production (*A*), rate of gas production (*c*), in vitro organic matter degradability (IVOMD), ammonia (NH_3_-N), and volatile fatty acids (VFAs).

	Alfalfa			Sainfoin			Sainfoin+PEG				*p*-Values		
Item	VEG	Start-F	End-F	VEG	Start-F	End-F	VEG	Start-F	End-F	r.s.d. ^2^	S	SM	SxSM
pH	6.42 ^a^	6.44 ^a^	6.41 ^a^	6.35 ^bx^	6.32 ^by^	6.33 ^by^	6.37 ^bx^	6.31 ^bz^	6.34 ^by^	0.032	<0.001	0.002	0.01
Gas production (mL/g dOM ^3^)	179	184	188	183	163	181	180	199	173	26.5	0.43	0.97	0.12
*A* (mL)	68 ^x^	67 ^bxy^	62 ^y^	63 ^y^	74 ^bx^	70 ^x^	65 ^y^	78 ^ax^	67 ^y^	5.7	0.01	<0.001	<0.001
*c* (h^−1^)	0.2^a^	0.22 ^a^	0.19	0.14 ^by^	0.16 ^bxy^	0.19 ^y^	0.15 ^bx^	0.15 ^bx^	0.19 ^y^	0.037	<0.001	0.02	0.05
CH_4_ production (mL/g dOM ^3^)	37	3	39	37	33	38	37	39	36	5.0	0.29	0.73	0.17
CH_4_: gas (mL/L)	145	142	141	141^xy^	134 ^y^	148 ^x^	145	144	144	8.8	0.35	0.13	0.11
IVOMD (g/kg)	851 ^bx^	837 ^bx^	770 ^by^	850 ^by^	926 ^ax^	856 ^ay^	868 ^ay^	936 ^ax^	864 ^ay^	16.9	<0.001	<0.001	<0.001
NH_3_-N (mg/L)	240 ^a^	247 ^a^	210	206 ^b^	184 ^b^	201	242 ^a^	234 ^a^	215	31.7	<0.001	0.06	0.20
Total VFAs (mmol/L)	102 ^xy^	105 ^x^	97 ^y^	96 ^y^	103 ^x^	101 ^xy^	99 ^y^	105 ^x^	102 ^xy^	7.1	0.61	0.02	0.35
Acetic acid (C_2_) (mol/100 mol)	63.1 ^cy^	63.9 ^bx^	64.0 ^bx^	65.4 ^ay^	66.5 ^ax^	66.3 ^ax^	63.9 ^by^	64.6 ^bx^	64.5 ^bx^	0.76	<0.001	<0.001	0.95
Propionic acid (C_3_) (mol/100 mol)	15.5 ^a^	15.3 ^a^	15.3 ^a^	15 ^bx^	14.5 ^cy^	14.5 ^by^	15.3 ^ax^	14.9 ^by^	15.2 ^ax^	0.28	<0.001	<0.001	0.09
Butyric acid (mol/100 mol)	13.0 ^a^	12.6 ^a^	12.6 ^a^	12.1 ^c^	12.2 ^b^	12 ^b^	12.5 ^b^	12.8 ^a^	12.4 ^a^	0.45	<0.001	0.20	0.16
Iso-butyric acid (mol/100 mol)	1.97 ^ax^	1.91 ^axy^	1.87 ^ay^	1.82 ^bx^	1.68 ^by^	1.72 ^by^	1.97 ^ax^	1.83 ^ay^	1.84 ^ay^	0.100	<0.001	<0.001	0.63
Valeric acid (mol/100 mol)	2.14 ^axy^	2.09 ^ay^	2.22 ^ax^	1.87 ^bx^	1.7 ^cy^	1.83 ^cx^	2.12 ^ay^	1.95 ^bx^	2.04 ^by^	0.107	<0.001	<0.001	0.11
Iso-valeric acid (mol/100 mol)	4.27 ^ax^	4.09 ^axy^	4.04 ^ay^	3.87 ^bx^	3.5 ^by^	3.65 ^by^	4.19 ^ax^	3.91 ^ay^	3.93 ^ay^	0.223	<0.001	<0.001	0.69
C_2_:C_3_ (mol/mol)	4.1 ^cy^	4.18 ^cxy^	4.2 ^bx^	4.39 ^ay^	4.62 ^ax^	4.59 ^ax^	4.18 ^by^	4.36 ^bx^	4.25 ^by^	0.109	<0.001	<0.001	0.13
CH_4_:VFA (mL/mol)	2.41	2.24 ^b^	2.35	2.53	2.22 ^b^	2.48	2.62	2.73 ^a^	2.4	0.313	0.01	0.26	0.10

^1^ VEG: vegetative; Start-F: start of flowering; End-F: end of flowering; ^2^ residual standard deviation; ^3^ degraded organic matter. Within a parameter, means with different superscript (a,b,c) differ at *p* < 0.05 for the substrate effect in each stage of maturity; with different superscripts (x,y,z) at *p* < 0.05 for the stage of maturity effect in each substrate.

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
