# Peer review of "Methane Production of Fresh Sainfoin, with or without PEG, and Fresh Alfalfa at Different Stages of Maturity is Similar but the Fermentation End Products Vary"

_animals, 2019, doi:10.3390/ani9050197_

Round 1
Reviewer 1 Report
To my opinion, this is a good paper, which provides important data. However, a major revision is required. I suggest the following for your revision:
Author affiliations: If you all have the same affiliation, I think you do not need the superscript following your names.
Please revise English language and style in the whole text. E.g., L 10: "increasing".
Simple summary: I miss a short statement on the main results and conclusions.
Abstract: L 27: check abbreviation "TC". L 35, 36: check syntax. What is the quotient telling us? Please explain. Did you found a reduction of methane production?
Keywords: please change "dry matter digestibility" to "in vitro organic matter digestibility".
Introduction: L 42: please explain this statement in a bit more detail. L 57: it is not necessary to explain chemical formulas used as abbreviation.
Materials and methods: L 78: "...described by ...". L 88: CP, and elsewere, check if all used abbreviations are explained when first mentioned.
IMPORTANT: Sections 2.2.2., 2.2.3., and 2.3.1.: please provide a much more detailed description of the used methods. You provide the references, but please also give us some brief descriptions. The methods should be reproducable without the need of extensive literature search. Was the ruminal fluid mixed from the four wethers? L 101: please give SD with one decimal place, and check this carefully throughout the manuscript. How did you produce the inoculum? Did you measure pH and redox potential of the ruminal fluid and the inoculum before fermentation? Please provide these data. Which settings did you use in the in vitro system, e.g., valve open time, threshold for gas release, agitation, etc. Which buffer did you use, and was it reliable for a 72 h fermentation? How did you receive solids for ash analysis and organic matter digestibility calculation (did you use filters, nylon bags, etc.)? Please give a more detailed explanation. It is not necessary to describe all nutrient analyses in detail, but please describe the analyses of the main determinants in your paper: methan (provide details on sampling and GC analysis including GC settings), VFA (information on preparation and GC analysis), NH3-N, and tannins. Please clearly state that you used aNDFom and ADFom. L 130, and other: check the manuscript for correct citation, and provide only the first author's name followed by et al.
L 141: did you also considered other models and did you check which is best fitting your data?
L 149: did you use several GCs? Please describe GC equipment, analysis, and settings in more detail.
Results: L 170: please explain "lignin (sa)". L 173: EE and NSC contents decreased in sainfoin.
Table 1: please explain all abbreviations in a footnote.
L 187: again, check if all abbreviations are correctly explained when first mentioned.
Section 3.2.: did you test and verify the success of PEG addition? Please provide data, and/or an explanation.
Figure 1: please give the used number of replicates per substrate and/or the degrees of freedom.
Table 2 and Table 3: should be combined to one table, and the data should be rearranged. In Table 3 LSM of substrates include the phenological stage, and LSM of phenological stage include all substrates. Is this correct? I would recommend to give LSM for each substrate at each phenological stage, and give the p-value for substrate, phenological stage, and the interaction, respectively. By the way, I think "phenological stage" should be better "stage of maturity" or the like. Again, please add explanations for the used abbreviations, numbers of replicates, or DF, and I would prefer that you give individual SE or a range of SE.
The Results section is very long and hard to read. I would suggest you to shorten it and summarize the results. It is not necessary to describe all I can see in the tables, only the main and in the context most important results.
Clearly indicate what was measured (gas production, methane, etc.), and what was calculated/estimated (A, c).
Please give p-values as follows: P > ... (with spaces).
Discussion: please start with a short introduction into this section. Remember the reader what are your objectives, and why this is important. Please add a short criticism of the methods.
L 289: did you consider the buffering capacity of the substrates. Try to give some data from the literature.
L 296: ... and of type and settings of the in vitro assay.
L 296-298: please explain that sentence.
L 306-307: explain why.
L 333: "... degradation of ...".
L 332-334/354-355: please briefly explain the mechanism.
L 363: give the author's name.
L 370, and others: if you use the Oxford comma for enumerations, please use it consistently throughout the text.
L 388, and others: please write "in vivo" italic as well.
L 393, and others: please use British or American spelling consistently, not a mix of both.
You give some correlations in the discussion, but they are not included in the Results section. Please add a table, or some sentences.
L 397-400: this needs to be explained. Isn't the measured gas production more reliable for the discussion here than the estimated (A)? Are you sure that the used model to calculate A is fitting?
L 402-404: explain why.
L 419-423: again, explain why.
Conclusions: L 425: delete "excellent". L 462: "N-NH3" ? L 427: "... valeric acid proportion from sainfoin in vitro." L 428: "It would be interesting ..." should read "It is required ...".
Please carefully check the references. E.g., 17.) Crop Science, 20.) Journal of Dairy Science, 22.) indent, 29.) Name of last author: F. Dohme-Meier ?
Author Response
The answer to reviewer 1 has been uploaded in word

Reviewer 2 Report
The paper is well written and the treated topics are interesting and taking in count one of the most innovative aspect of livestock nutrition. Consequently I suggest it for the publication after minor revisions.
Some suggestion were reported into the attached file. I suggest to the authors to check the guidline for the authors because into the text the references were not alwais reported following the guidelines.

Author Response
The answer to reviewer 2 has been uploaded in word

Round 2
Reviewer 1 Report
Dear authors. You provided a convincing revision. I have only some minor suggestions.
Please try to give a short conclusion in the simple summary. I know there is not much space.
Please give the ruminal fluid pH before incubation in materials and methods.
Please carefully check American or British spelling, i.e., fiber or fibre.
Please decide if you want to use "phenological stage" or "stage of maturity". I found both versions in the text.
Good job. Thank you.
Author Response
The comments for Reviewer 1-Round 2 has been uploaded in word document. The new chages has been provided in red.
